# Spectral Indices as a Tool to Assess the Moisture Status of Forest Habitats

Adam Młynarczyk [1,*], Monika Konatowska [1,2], Sławomir Królewicz [1], Paweł Rutkowski [2], Jan Piekarczyk [1] and Wojciech Kowalewski [3]

1   Environmental Remote Sensing and Soil Science Research Unit, Faculty of Geographic and Geological Sciences, Adam Mickiewicz University in Poznań, Wieniawskiego 1, 61-712 Poznań, Poland
2   Department of Botany and Forest Habitats, Faculty of Forestry and Wood Technology, Poznań University of Life Sciences, Wojska Polskiego 71F, 60-625 Poznań, Poland
3   Department of Artificial Intelligence, Faculty of Mathematics and Computer Science, Adam Mickiewicz University in Poznań, Wieniawskiego 1, 61-712 Poznań, Poland
*   Correspondence: adam.mlynarczyk@amu.edu.pl

**Abstract:** Measurement of water content in forest habitats is considered essential in ecological research on forests, climate change, or forest management. In the traditional forest habitat classification, two systems of habitat conditions analysis are found: single factor and multifactor methods. Both are laborious and therefore costly. Remote sensing methods provide a low-cost alternative. The aim of the presented study was to find the relationship between the spectral indices obtained from satellite images and the forest habitats moisture indices used traditionally in the Polish forest habitats classification. The scientific hypothesis of the research is as follows: it is possible to assess the variation in the humidity of forest habitats on the basis of spectral indices. Using advanced geographic information system (GIS) technology, 923 research plots were tested, where habitat studies performed with the traditional methods were compared with the analysis of 191 spectral indices calculated for Sentinel-2 image data. The normalized difference vegetation index (NDVI) has proved to be the most useful to the assessing of moisture of forest habitats. The ranking of the most correlated indices was calculated as $E_{intg}$—the product of the absolute value of the slope and the mean square error complement, and for the top five indices was as follows: NDVI = 0.248619, EXG = 0.242112, OSAVI = 0.239412, DSWI-4 = 0.238784, and RDVI = 0.236995. The results also highlight the impact of water reservoirs on the humidity and trophicity of forest habitats, showing a decrease in the fertility of habitats with an increase in distance from the water reservoir. The results of the study can be used to preparing maps of the diversity of forest types, especially in hard-to-reach places, as well as to assess changes in the moisture status of habitats, which may be useful, for example, in the assessment of the fire risk of forest habitats. We have proved that NDVI can be used in applications for which it was not originally designed.

**Keywords:** NDVI; forest typology; forest humidity; forest trophicity; Sentinel-2

## 1. Introduction

Forests play a key role in the Earth's water cycle [1,2] and therefore the measurement of water content in forest habitats is considered essential in most ecological research on forests, climate change or forest management, as well as against floods, fires, and soil erosion. In traditional forest habitat classification, two systems for the analysis of habitat conditions are found: single factor or multifactor methods [3]. Single factor methods rely on one factor to describe a forest site, such as soil or climate, whereas multifactor methods are based on interrelationships between climate, physiography, soil, and vegetation [4]. An example of the use of the multifactor factor method is the digital site classification maps used in Germany, containing information on the soil properties, as well as on climatological and topographic factors [5]. The similar multifactor method is used in Poland [6,7]. It is based

on the recognition of the features of tree stands, plant communities, and soil, combining these elements into one abstract unit—the forest habitat type (FHT). In total, there are 38 FHT units in Poland, divided on the basis of their geographic location into lowland (15), highland (8), and mountain (15) units. The basic element of the assessment of forest soils is their trophic and moisture differentiation. Due to the trophic diversity of forest habitats, the following ones can be distinguished in Poland: coniferous forests, conifer-dominated mixed forests, deciduous-dominated mixed forests, and deciduous forests. With regard to humidity diversity, the FHTs are divided into dry, mesic, moist, and swampy. The combination of trophic and moisture features results in the name FHT. For example, if a forest trophically belongs to the group of conifer-dominated mixed forests and in terms of humidity to mesic types, then the forest habitat type is called a mesic conifer-dominated mixed forest.

The criteria for the differentiation of FHTs into moisture groups are based on the assessment of two basic types of water: habitats depending on precipitation and habitats depending on groundwater. The forest habitats moisture index (FHMI), based on groundwater level, is assessed depending on the depth of water in the soil. Habitats with water on the ground surface or at the depth of 0–20 cm are described as "g1", habitats with groundwater in the range of 20–50 cm as "g2", in the range of 50–80 cm as "g3", in the range of 80–180 cm as "g4" and in the range of 180–250 cm as "g5". When the water is below 250 cm, the habitat is classified as "g6", and when the habitats are additionally covered with the poorest forms of pine forests in Poland, the so called "Dry Forest", they are classified as "g7" [6,7].

Studies that analyze the FHMI in this way are carried out in Poland on all lands managed by the State Forests National Forest Holding, covering approx. A total of 23% of the country's land area and approx. 77% of the Polish total forest area [8], with an accuracy of one research plot per 4–12 ha, depending on the diversity of geology and geomorphology of the terrain. They are repeated every 30 years. This method of assessing the suitability of forest soils in terms of forest management is quite precise, but laborious and costly. Therefore, faster and cheaper methods are sought, including those based on remote sensing [9–12].

In numerous papers on the use of remote sensing methods, soil moisture is analyzed mainly in its surface layers, e.g., [13–16], while the general water supply is important for the development of the forest, including soil moisture, precipitations, and air humidity. Therefore, the main assumption of the study was to look for indirect methods in assessing the FHMI, based on plant indicators (NDVI and others), assuming that remote sensing methods can indicate such a condition of forest vegetation that reflects the diverse moisture sources of forest habitats.

Remote sensing methods based on the registration of spectral reflectance have been used for several decades and, similarly to terrestrial techniques, are developed mainly for the needs of agriculture [17–22]. Remote sensing methods are also used in forestry, but mainly for assessing the health condition of the forest, as well as abiotic threats, such as fires [23] or droughts (e.g., [24]), and biotic ones, such as insect outbreaks [25].

Remote sensing vegetation studies are based on near and far infrared radiation as well as on other ranges of radiation reflected and absorbed by plants (e.g., [26–29]). Remote sensing uses aerial, ground, and satellite imagery. Despite the dynamic development of measurement techniques (e.g., [30–36]) in attempts to study the FHMI on the basis of remote sensing, the obstacle is the forest's key element—the plant cover [37]. So far, a barrier to the use of available remote measurement methods is also the low resolution of the image products of such studies [16,38].

Studies on the possibility of using remote sensing in estimating the moisture content of forest soils are carried out in many places around the world. Nijland et al. [39] investigated the relationship between soil moisture and site productivity of four types of Canadian forests (CD: conifer-dominated; MX: mixed conifer–deciduous; DU: deciduous-dominated with conifer understory; DD: deciduous-dominated) and NDVI maximum pre-harvest

values from Landsat images. The authors showed that the relationship between NDVI and the humidity of the habitat was negative and the relationship with the depth of groundwater was positive. Stronger correlation between productivity and moisture content was observed in coniferous and mixed stands, compared to two types of forests with a predominance of deciduous-dominated forest types. Moreover, the habitats of mixed and coniferous forests were characterized by greater diversity in both humidity and productivity than habitats with a predominance of deciduous trees.

In forests of the German Rhineland-Palatinate state, Dotzler et al. [40] conducted research on detecting tree water stress responses in deciduous forests using hyperspectral aerial images. The spectral index PRI (photochemical reflectance index) calculated on their basis showed differences between habitats caused by drought and depending on soil moisture conditions.

Coniferous species (mainly *Pinus sylvestris*) dominate in Polish forests {68.2% of the forest area, including pine, which covers 58% of all forest areas, 60.1% of the area of the State Forests, and 54.5% of private forests [8]}. The dominant type of forest soils in Poland are rusty soils [41]. With regard to forests in Poland, the main sources of water are atmospheric precipitation and air humidity [42].

The aim of this study was to find the relationship between the spectral indices obtained from satellite images and the moisture of forest habitats, allowing for their practical use in forest management.

The scientific hypothesis is as follows: it is possible to assess the variation in the humidity of forest habitats on the basis of spectral indices.

## 2. Materials and Methods

### 2.1. Research Area

The research area is the Bory Tucholskie National Park (BTNP), considered as a representative area for Polish forest conditions, located in northern part of Poland (Figure 1), in formerly glaciated areas with a varied topography, cut by the gutters of glacial lakes. The altitude above sea level ranges from 120 to 140 m (Figure 1C). Over 90% of BTNP is covered by sub-oceanic pine forests (*Leucobryo-Pinetum* plant association) [43]. The dominant type of soil are rusty soils, formed mainly of outwash sands. The extreme points of research area border are marked by geographical coordinates: west 17°30′08″ E, 53°48′54″ N; north 17°32′30″ E, 53°51′23″ N; east 17°37′06″ E, 53°47′58″ N; south 17°34′51″ E, 53°46′15″ N.

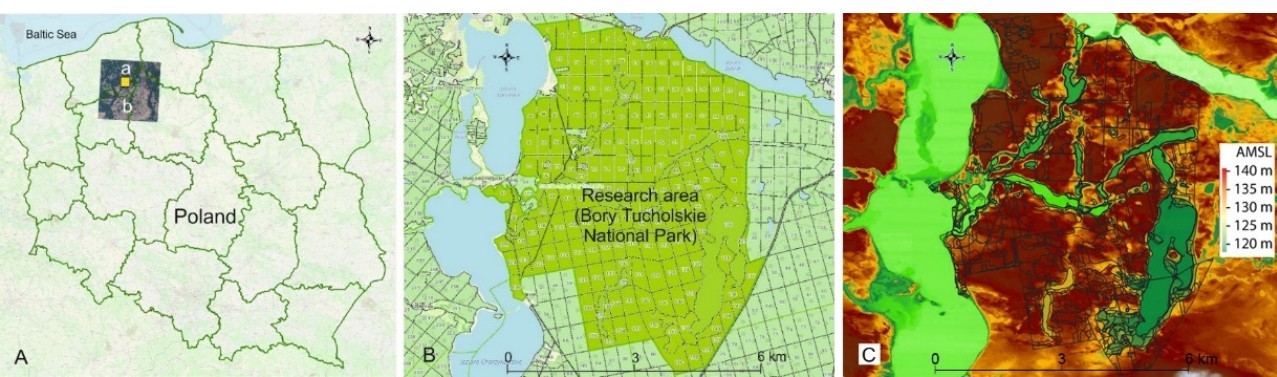

**Figure 1.** Location of the research area in northern Poland (yellow square, (**A**)-a; (**A**)-b: satellite image on the background of the map of Poland used in the research; (**B**)—the area of BTNP, (**C**)—elevation map of research area.

The soil cover of the Bory Tucholskie National Park shows relatively little typological variation, which results from the relatively homogeneous geological structure of the substrate. The parent materials of the soils are mainly sandy glacial sediments of the Vistula glaciation and the Holocene, most often loose sands, in total, covering 98.6% of the park's area. Under the dominance of sandy parent materials, characterized of low water

retention, forests are natural vegetation. In such conditions, mainly podzolic-rusty soils have developed, less often typical podzolic soils or brunic rusty soils. These three soil subtypes cover a total of 3706 ha and constitute 80% of the Park's area.

## 2.2. Satellite Data

The search and download of Sentinel-2 (A,B) images of the European Space Agency (ESA) was accomplished using the Polish data repository of the Copernicus program (https://www.copernicus.eu/en, accessed on 27 July 2022), Sat4Envi (https://sat4envi.imgw.pl/, accessed on 27 July 2022) managed by the Polish Institute of Meteorology and Water Management (IMGW-PIB). The characteristics of the Multispectral Instrument (MSI) sensor on S2A and S2B platforms are available on the ESA website (https://sentinels.copernicus.eu/web/sentinel/user-guides/sentinel-2-msi, accessed on 27 July 2022). Data from the level of the product Level L2A were used for the analysis. L2A means that each image pixel for spectral band contains calibrated reflectance at earth surface. It is a product that was created as a result of geometric correction, taking into account the influence of topography on the image and radiometric correction of radiation changes in the atmosphere [44,45].

As shown in Table 1, the differences between Sentinel 2A and 2B are insignificant and were considered as insignificant from the point of view of the conducted studies.

**Table 1.** Most frequent Sentinel 2 channels used in the calculations [44].

| Sentinel 2 bands | B2 | B3 | B4 | B8 |
|---|---|---|---|---|
| Spatial resolution (m) | 10 | 10 | 10 | 10 |
| Sentinel 2A central wavelength (nm) | 496.6 | 560.0 | 664.5 | 835.1 |
| Sentinel 2B central wavelength (nm) | 492.1 | 559.0 | 665.0 | 833.0 |
| Sentinel 2A bandwidth (nm) | 98 | 45.0 | 38.0 | 145.0 |
| Sentinel 2B bandwidth (nm) | 98 | 46.0 | 39.0 | 133.0 |
| | pigment chlorophyll absorptions in blue [46] | pigment chlorophyll minimum absorption in green band [46] | pigment chlorophyll absorptions in red band [47] | responsive to canopy structural variations, canopy type and architecture [47] |

The BTNP is located in the area of one section (Tile) (Figure 1A-b) designated T33UXV (UTM). The period of image search was limited to three years, from 1 January 2018 to 31 December 2020 and the criterion of cloud cover covering the section was less than or equal to 70%. Such a large value of the cloudiness parameter was assumed, since the park covers only 0.38% of the section area (whole park area 46.13 km$^2$/Tile area 100 × 100 km = 10,000 km$^2$). Then, the acquired images were cropped to a regular fragment including the BTNP using a geometric object in SHP format. Cropped images were done in the TNTmips version 2022 software from Landscan (US, San Luis Obispo, CA, local license for Adam Mickiewicz University). In the next step, images with no cloud cover in the research area were selected. The initial image classification layer SCL (Scene Classification Layer) was used for this purpose. It was checked as to whether there are values on this layer corresponding to clouds and their shadows, as well as snow cover. In the SCL classification raster, such surface categories are marked with the following values: shadows—3, clouds with low probability—7, medium—8, high—9, cirrus clouds—10, snow—11. Finally, the criterion eliminating the display term defined as follows was applied: if

there were clouds of medium (8) and high probability (9) in the polygon area and if the number of such pixels was greater than 50 in the image, such an image was excluded from the analysis; if the number was lower than 50 pixels, the spatial distribution of clouds and over which polygons they are located were visually checked. In the case of high scattering of individual pixels, the term of imaging was included in the set of analyzed images.

Geometric data on the ranges of forest habitat units (Figure 2), available in the SHP format, together with the database in the following formats: .prj, .sbn, .sbx, and .shx, were obtained from the website of the Bory Tucholskie National Park Geoportal (http://gis.pnbt.com.pl/, accessed on 27 July 2022). All data were recorded in the form of Attribute Tables saved in ArcGIS as ".dbf file". ArcGIS is software that enables to store, manage, and retrieve data and is used by authors according to Licence Agreement (E203 04/24/2012) between the Poznań University of Life Sciences and Environmental System Research Institute, Inc. ("ESRI"). Due to the spatial resolution of satellite images (10–60 m), units with an area smaller than 1 ha were excluded from the analysis, which also corresponds to the methodology of identifying forest habitat conditions using the traditional terrestrial method [7].

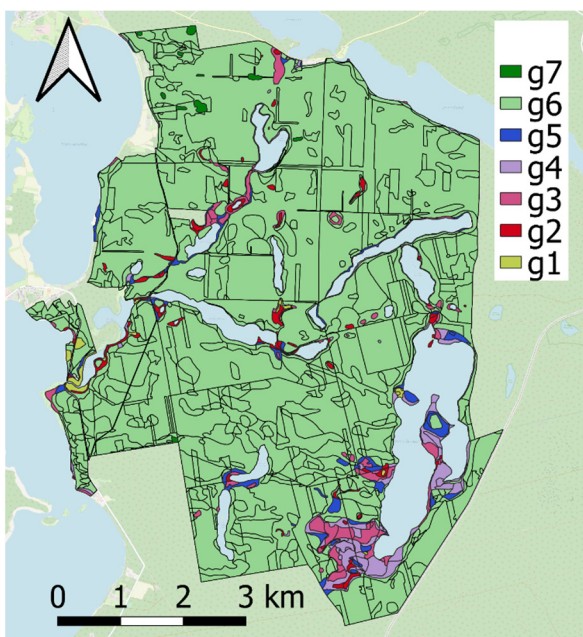

**Figure 2.** Map of humidity variation in Bory Tucholskie National Park, expressed in FHMIs (g1–g7) used in Poland. The explanation of g1–g7 indexes is given in the Introduction.

In the next step, for each Sentinel-2 data recording date, the average values from pixels DNs corresponding to a reflectance multiplication by 10,000, were calculated for forest habitat units (polygons)—for all spectral bands (except for band number 9, which is not present in the L2A product). Calculations were made in the TNTmips version 2022 software from Landscan (US, San Luis Obispo, CA, local license for Adam Mickiewicz University). The principle of calculating the mean of pixels values entirely located within the area of the forest unit was resorted to. Therefore, border pixels lying under the boundaries of forest habitat units were rejected from calculation.

In the next stage, for each image recording date, on the basis of the average DN values of the pixels of images from different spectral channels of the MSI Sentinel-2A sensor within the polygons of individual FHMIs, 249 vegetation indices were calculated, the formulas of which are stored and described in the "Index Database" (https://www.indexdatabase. de/, accessed on 27 July 2022) [48]. Ultimately, after the selection and analysis of the formulas, 191 indices were selected for the calculation. A list of all selected indices and their formulas (via a link to the appropriate page in the "Index Database") is included in the Supplementary Materials.

All further calculations were performed with the R Studio and visualized in Excel spreadsheet software from Microsoft. For all polygons belonging to the same humidity category, for a given image recording date, the mean value of the index ($M_{inTgi}$) were calculated. Then all index values ($M_{inTgi}$) were standardized according to the formula:

$$SM_{inTgi} = \frac{(M_{inTgi} - \overline{M_{inTgi}})}{\delta_{inTgi}}$$

where $SM_{inTgi}$—standardized mean value, $M_{inTgi}$—non-standardized mean value, and $\delta_{inTgi}$—standard deviation. The $SM_{inTgi}$ were grouped and averaged for all g1–g7 moisture indices. A linear regression was calculated from the grouped indices and the regression slope $aSM_{inTgi}$ was determined. In the next step, the total mean square error was calculated for each index—$MSE\_M_{inTgi}$. The ranking of the most correlated indices was calculated on the basis of $E_{inTg}$, meaning the product of the absolute value of the slope $aSM_{inTg}$ and the mean square error complement ($1 - MSE\_M_{inTgi}$).

The cross-validation method was applied to evaluate regression model for NDVI (0.017). This was done by repeating the k-fold cross validation procedure multiple times for different k and reporting the mean result across all folds from all runs.

## 3. Results

In the vector layer of the map of the park's habitats, 923 research plots were distinguished (Table 2), covering the area of forest habitats with seven degrees of FHMIs. In order to prevent the model from overfitting, the number of samples with different groups has been equalized, primarily by "g6" group's reduction. Within the "g1" FHMI, there were only peat soils, mostly fertile peats. In the "g2" and "g3" FHMIs, raised bogs and fen peat soils predominated. To the "g4" FHMI, groundwater gleyed soils have the greatest share, to the "g5", FHMI typical podzolic soils, to the g6, FHMI rusty podzolic soils, and to the g7, FHMI arenosols.

**Table 2.** Groundwater levels in the FHTs area of the Bory Tucholskie National Park.

| FHMI | Moisture Group of Habitats | Groundwater Table [m] | Number of Plots | Area [ha] |
|---|---|---|---|---|
| g1 | Swampy | 0.0–0.2 | 13 | 37.89 |
| g2 | Swampy | 0.2–0.5 | 70 | 121.44 |
| g3 | Swamy/Moist | 0.5–0.8 | 73 | 190.43 |
| g4 | Moist | 0.8–1.8 | 74 | 118.71 |
| g5 | Mesic | Below 1.8 | 58 | 170.64 |
| g6 | Mesic | Below 2.5 | 625 | 3653.34 |
| g7 | Dry | Below 2.5 | 10 | 17.14 |
| | Total | | 923 | 4309.59 |

Figure 3 shows the spectral characteristics of habitats with different indices of soil moisture obtained from the Sentinel-2 image recorded on 28 June 2021. Based on the level of the spectra (reflectance) in the near infrared range (channel 8A), three groups of habitats with different FHMI can be distinguished: (i) wet—"g1", with mainly deciduous trees, (ii) strongly moist—"g2", very moist—"g3", and moist—"g4", covered with mixed forest and (iii) moist-mesic—"g5", mesic—"g6", and dry—"g7", covered with pine forests. The driest habitats were distinguished by higher reflection in the visible range, especially in the red channel and in the range of short-wave infrared SWIR in channel 12.

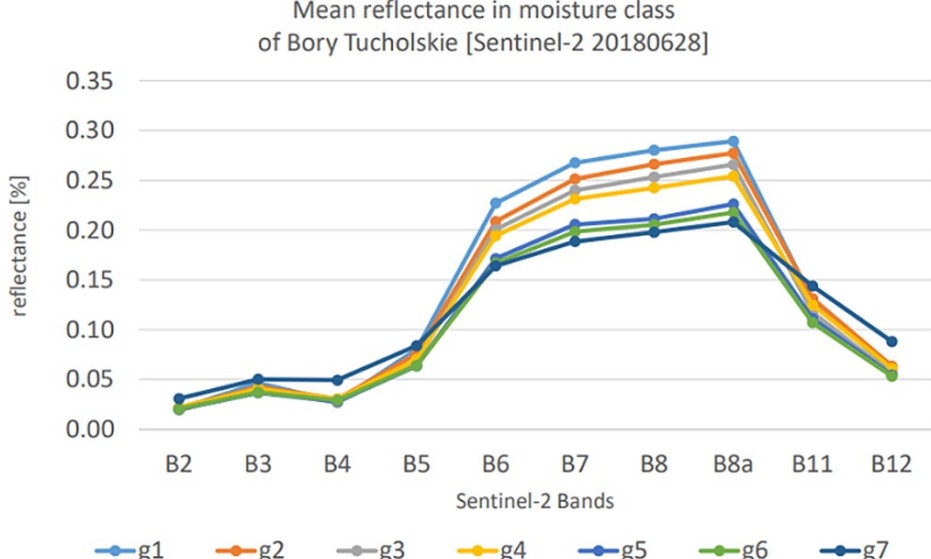

**Figure 3.** Spectra of seven FHMI in Bory Tucholskie National Park based on the Sentinel-2 image recorded on 28 June 2018.

Among the analyzed vegetation indices, NDVI turned out to be the most strongly correlated with FHMI, during the full growing season, which in Poland falls on the period from June to August.

Table 3 shows the top five indices along with $aSM_{inTg}$, $MSE\_M_{inTgi}$, and their ratio $E_{inTg}$.

**Table 3.** Ranking of the top five indices.

| Index | $aSM_{inTgi}$ | $MSE\_M_{inTgi}$ | $E_{inTg}$ | Formula | Citation |
|---|---|---|---|---|---|
| NDVI | 0.286434 | 0.132020 | 0.248619 | $NDVI = \frac{B8 - B4}{B8 + B4}$ | [49] |
| EXG | 0.282297 | 0.142348 | 0.242112 | $EXG = 2*B3 - B2 - B4$ | [50] |
| OSAVI | 0.278790 | 0.141247 | 0.239412 | $OSAVI = (1 + 0.16)\frac{B8 - B4}{B8 + B4 + 0.16}$ | [51] |
| DSWI-4 | 0.274647 | 0.130578 | 0.238784 | $DSWI - 4 = \frac{B3}{B4}$ | [52] |
| RDVI | 0.274793 | 0.137549 | 0.236995 | $RDVI = \frac{B8 - B4}{(B8 + B4)^{0.5}}$ | [53] |

Figures 4–6 show the variation of the values of the vegetation indices, calculated for the relationship between the FHMIs and the average values of the NDVI obtained from satellite images recorded in 2018, 2019, and 2020. Analyzing the data on the graphs, it should be noted that the degrees of moisture content correlated with the NDVI do not represent a specific depth of groundwater, but the ranges in which the groundwater level fluctuates during the growing season. It should also be mentioned that the boundaries of the groundwater level depth ranges are tangent for the adjacent FHMIs. The NDVI values are also in such a sequence, showing similar values for the neighboring moisture indices and clearly differentiating between the groups of moisture in forest habitats: swampy (g1–g3), moist (g3–g4), mesic (g5–g6), and dry (g7), as shown in Figure 4a, also distinguishing indexes g6 and g7, assuming the lowest NDVI values and representing soils with no groundwater at a depth of up to 250 cm.

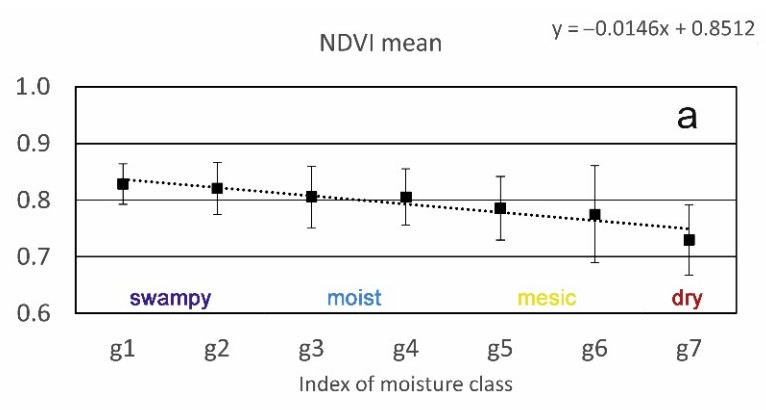

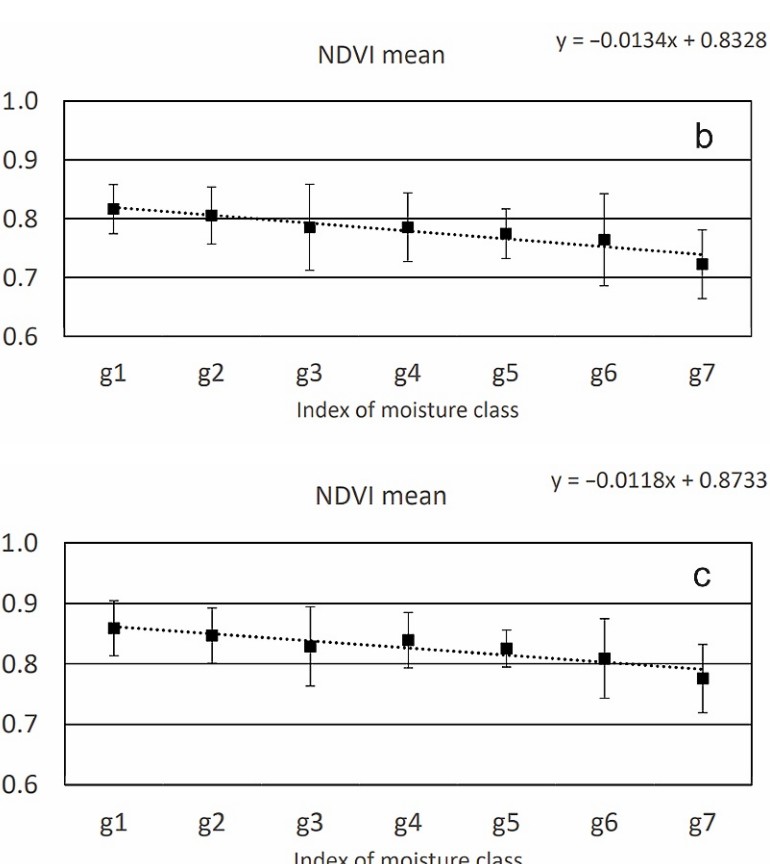

**Figure 4.** (**a**) Mean NDVI value from the Sentinel 2 image obtained on 20 July 2018, taking into account all FHMIs (g1–g7). (**b**) Mean NDVI value from the Sentinel 2 image obtained on 30 July 2019, taking into account all FHMIs (g1–g7). (**c**) Mean NDVI value from the Sentinel 2 image obtained on 6 August 2020, taking into account all FHMIs (g1–g7).

When selecting the dates for the data analysis in the years 2018–2020, with the aim of preparing the charts presented in Figures 4–6, an attempt was made to select the most similar dates in the individual years. In 2018 and 2019, for the presented examples, it was July 20. In 2020, due to weather conditions, good-quality satellite images were not obtained until August 8. Hence the discrepancy in the dates.

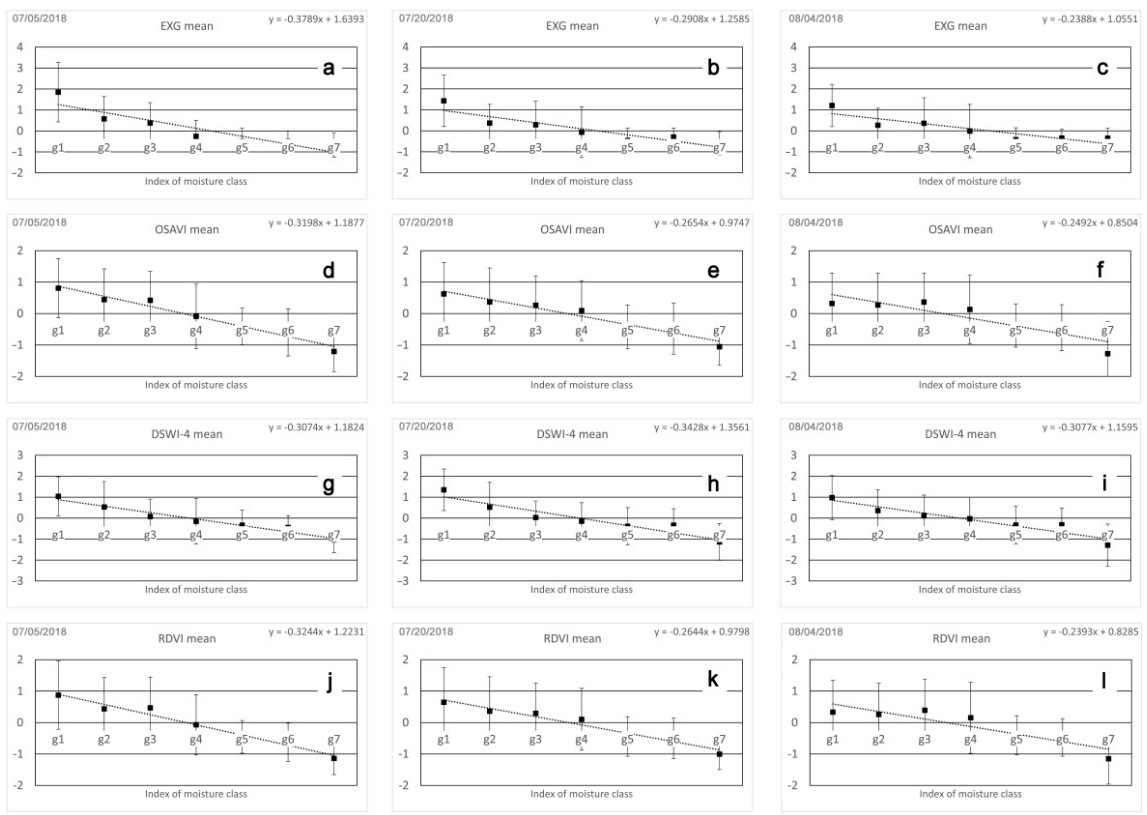

**Figure 5.** Average value of top indies in seven FHMIs: (**a**) EXG on 5 July 2018; (**b**) EXG on 20 July 2018; (**c**) EXG on 4 August 2018; (**d**) OSAVI on 5 July 2018; (**e**) OSAVI on 20 July 2018; (**f**) OSAVI on 4 August 2018; (**g**) DSWI-4 on 5 July 2018; (**h**) DSWI-4 on 20 July 2018; (**i**) DSWI-4 on 4 August 2018; (**j**) RDVI on 5 July 2018; (**k**) RDVI on 20 July 2018; (**l**) RDVI on 4 August 2018.

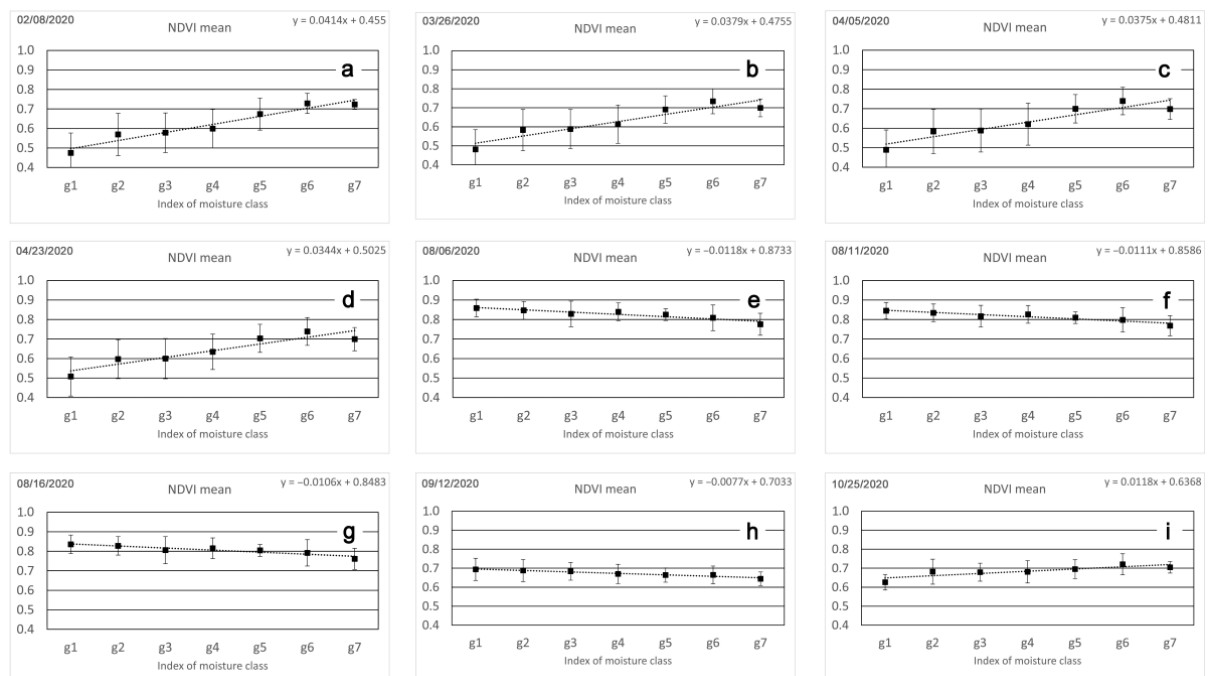

**Figure 6.** Average NDVI value of the index over year 2020: (**a**) on 8 February; (**b**) on 26 March; (**c**) on 5 April; (**d**) on 23 April; (**e**) on 6 August; (**f**) on 11 August; (**g**) on 16 August; (**h**) on 12 September; (**i**) on 25 October.

Figure 5 shows example dates with the highest indices positions in the ranking (Table 3). A ranking of all indices, listed from the best to the worst correlated with the FHMIs, can be found in the Supplementary Materials.

Figure 6 shows changes in the relationship between NDVI and FHMI at different dates in the same year (2020). This analysis illustratively shows the relationship between NDVI, the FHMIs, and the seasons of the year. Lack of data for g1–g3 FHMIs and low NDVI value for g4 FHMI results from the trophic nature of wet habitats, which, as generally more fertile, have a greater share of deciduous species. In winter and early spring there are no leaves on these trees in Poland, while habitats g5 and g6 are mostly and the g7 habitats fully associated with pine, which maintains the assimilation apparatus throughout the year. Based on the graphs shown in Figure 6, it can be assumed that in spring and autumn the trend line is ascending (no leaves on deciduous trees, presence of pine needles), in summer the trend line is descending (deciduous trees have more chlorophyll, with full foliage). In winter, when the plants are dormant, the NDVI values are relatively low (below 0.6) for obvious reasons.

As has been shown and as mentioned earlier, the NDVI correlates with the FHMIs in the summer and this period should be taken into account in Central Europe when determining the degree of moisture in forest habitats on the basis of NDVI.

In order to illustrate the relationship between NDVI and FHMIs, the images of site maps made with the traditional method of soil pits were also compared with the image of the same area differentiated according to the NDVI values (Figure 7), grouping swampy (g1–g3 FHMIs), moist (g4), moist-mesic (g5), and mesic habitats (g6).

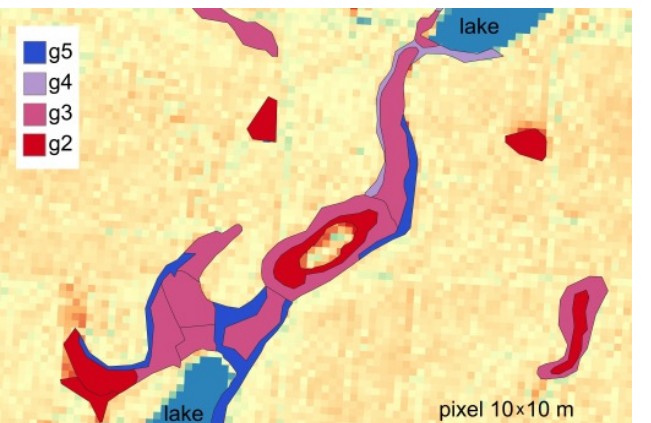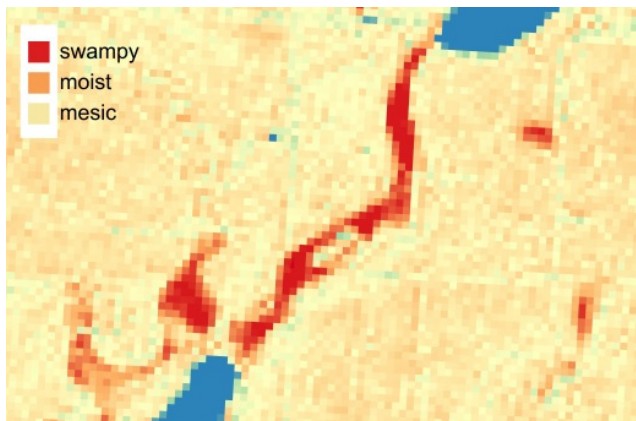

**Figure 7.** Comparison of the image of a fragment of the habitat map made with the traditional method of soil pits on the background of NDVI map (**left**) with the image of the same area differentiated due to the NDVI values (**right**). Only g2–g5 FHMIs are included in the image legend on the left. The main background of the picture shows the g6 FHMI. The g1 and g7 FHMIs do not exist in the presented part of area.

It should be noted, however, that apart from NDVI, also EVI, SAVI, SR520/670 and OSAVI indexes had relatively high compliance with the humidity of forest habitats, but these indicators also reflect on areas with little vegetation cover, such as forest roads or stands up to 15 years old (Figure 8).

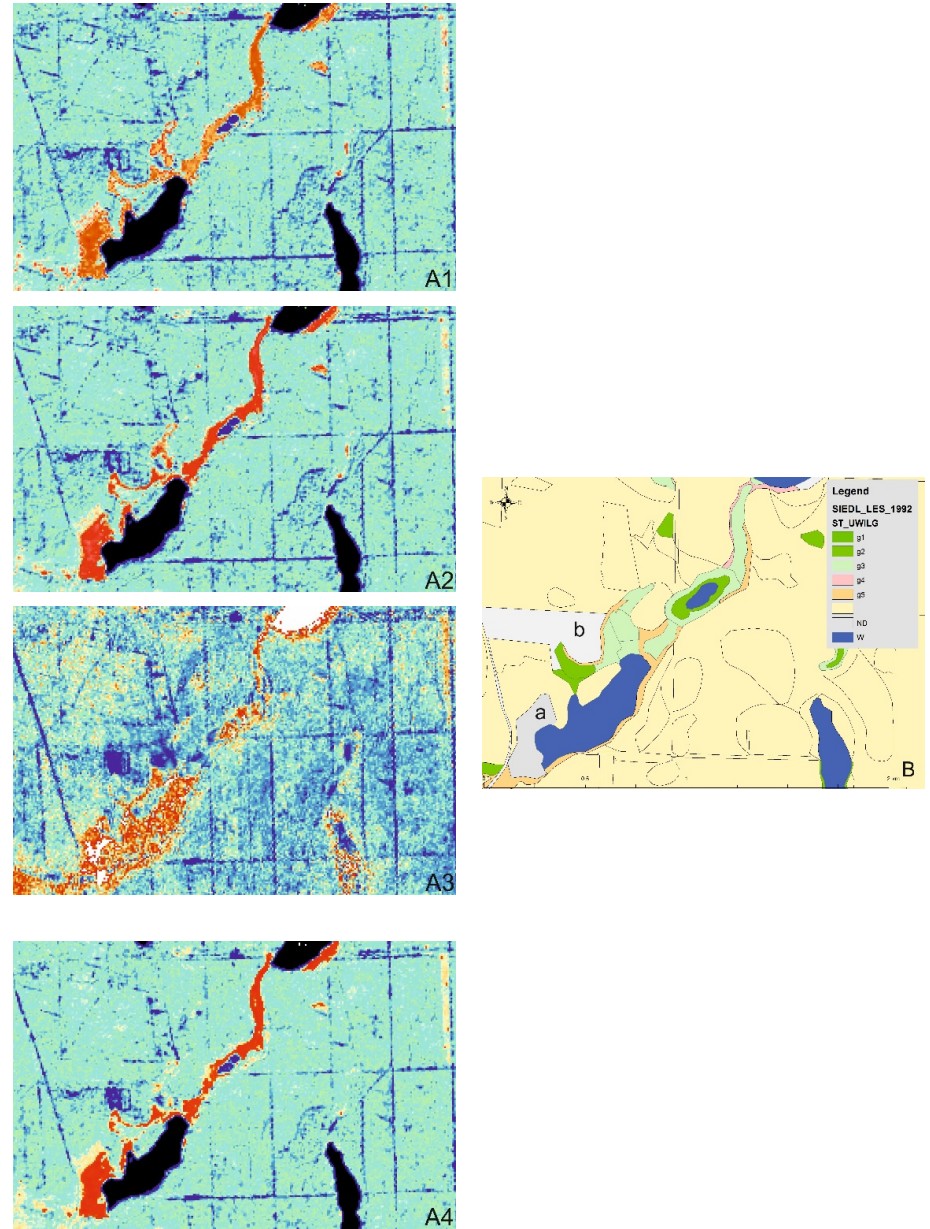

**Figure 8.** Comparisons of EVI map (**A1**), SAVI map (**A2**), SR520/670 map (**A3**), OSAVI map (**A4**) with the moisture content map of BTNP (**B**). The letter "a" on the B map indicates a meadow, the letter "b" the private land taken over by the park, for which no habitat development has been carried out, hence there is no data on the degree of moisture content. FHMIs g1 and g2 were combined into one green color.

Finally, the trophic system of forest habitats of the BTNP fragment was compared with the image obtained on the basis of the NDVI. Figure 9B clearly shows the zone of more fertile habitats (red), concentrated on the edges of the lakes and the declining fertility of the habitats to the east from the largest lake visible in the figure. It should be added that westerly winds prevail in Poland, which means that humid air masses from the lake move mainly to the east, and this is also the habitat fertility gradient shown in Figure 9A,B. Although there is also a wide zone of more fertile habitats to the west of the lake, it is associated with the rivers that flow into the lake in these places. In contrast, to the east of the lake, the habitat fertility gradient can be attributed to air humidity, which decreases with distance from the lake. It should also be added that, as indicated in Figure 9A,B, the habitat fertility gradient change zone is associated with the same type of soil (rusty soils),

which, depending on the vegetation cover, can be differentiated into podzolic rusty (RDb), typical rusty (RDw) and brunic rusty (RDbr) soils—Figure 10.

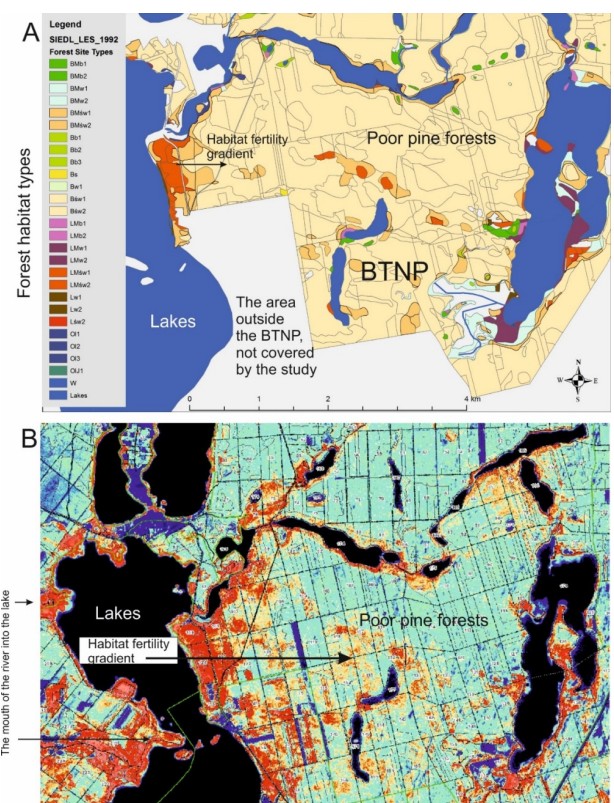

**Figure 9.** Variation of the trophic nature of forest habitats according to the map of forest habitats prepared using the traditional method (**A**) and on the basis of the NDVI (**B**). Red color in (**B**) correlates to zone of more fertile habitats presented in (**A**) (LMśw—deciduous-coniferous mixed forests; BMśw—coniferous-deciduous mixed forests).

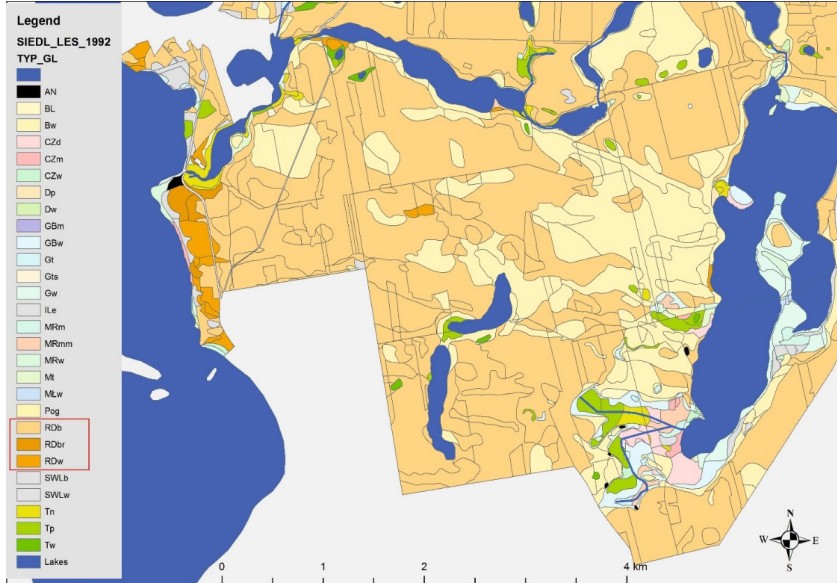

**Figure 10.** Classification of the soils of the selected area of the BTNP (according to http://gis.pnbt. com.pl/, accessed on 27 July 2022) on the basis of traditional soil pits. In the legend, rusty soils are marked with a red rectangle, differentiating into three subtypes: rusty podzolic (RDb), typical rusty (RDw), and brunic rusty (RDbr). Podzolic soils (BL and Bw) also have a significant share on the map.

## 4. Discussion

The presented data indicate a relationship between NDVI and the Forest Habitat Moisture Indexes, which is broader than the relationship between NDVI and soil moisture presented by Huete [54] in the results of his work as early as 1988. However, the author considered this relationship between NDVI and soil condition as a kind of obstacle in the NDVI analysis and paid attention to dynamic changes in the condition of soils related to their moisture content. The conducted research indicates that it is rather an advantage in understanding the spectral properties of soils and their interaction with vegetation and hydrological conditions.

Based on the conducted research, it is considered that the correct way to use NDVI to assess FHMIs should be to combine the degree of moisture into the following humidity groups: swampy, moist, mesic, and dry (Figure 4a). Such a connection may eliminate some discrepancies between the data obtained from the traditional identification of habitat conditions and the data obtained from satellite images. In the traditional method used in Poland [7], FHMIs are determined according to the current water level in the ground on the date of the research. The remote sensing method allows for the capture of differences in the moisture status of habitats in a long-term perspective. It should also be noted that the traditional method is based on the results of measuring groundwater at the test point, which is a soil pit representing an area of one to more hectares. Within the distinguished habitat contour, however, there may be differences in the moisture conditions, which is illustrated by satellite images with a resolution of $10 \times 10$ m. The obtained results from the analysis of remote sensing data may also reflect not only the influence of groundwater, but also the effect of other water sources, to which plants react, e.g., related to differences in air humidity. For the area where the research was conducted, which was abundant in lakes, this may be of significance, as pointed out by Rutkowski [42] when analyzing the diversity of humidity conditions in forest habitats for the area of Wielkopolska Region (the central-western part of Poland). On this basis, it can be concluded that climatic conditions are the key factor in the habitat of terrestrial ecosystems, which is also confirmed by the results of Rutkowski's [55] research. In this context, satellite images, reflecting the comprehensive impact of waters of various origins on forest ecosystems, may be a much more sensitive tool for assessing the moisture diversity of forest habitats than traditional methods based mainly on the description and assessment of soil diversity.

It should be noted that the concept of the "forest habitat moisture index" used in Poland does not only mean the presence of groundwater at a specific depth. Furthermore, in some regions of the world this information is unavailable. So, looking for the possibility of mapping groundwater-depended ecosystems other methods are used as a surrogate hydrological indicator [56–58]. One of them is FHMI. The concept of FHMIs reflects the total impact of water available from various sources of the forest ecosystem, with traditional, terrestrial measurement methods, which are often difficult to measure. With the dominance of sandy deposits in forest soils in Poland, where groundwater is generally below the root zone of trees, precipitation and water contained in moist air masses can be considered the main source of water. The impact of atmospheric precipitation on the forest is reflected in many papers (e.g., [59–61]), while the impact of air humidity related to water reservoirs on the development of forest ecosystems is poorly described, although the data collected in this study suggest that it is important. This is reflected, inter alia, in Figure 9.

## 5. Conclusions

The normalized difference vegetation index (NDVI) is the index widely used in remote sensing observations of live plants but it can be used in applications for which it was not originally designed. After analyzing 190 other indicators used in remote sensing, it was found that NDVI is very useful to assess the degree of forest habitats moisture, but the data must be obtained during the full growing season, which in the conditions of Central Europe (and the tested area) falls from June to August. The use of NDVI made it possible to distinguish and visualize on the map of the studied area four humidity groups of habitats

(swampy, moist, mesic and dry), coinciding with the results obtained using traditional, ground-based measurements.

The obtained results also indicate the possibility of using NDVI to assess the fertility gradient of forest habitats. Within the studied area it has been shown that the fertility of habitats decreases with increase in distance from the water reservoir.

The presented results of the study could be implemented in forestry practice, which should significantly reduce the costs of identifying the diversity of forest habitats. The usefulness of the obtained results may, for example, refer to the determination of the boundaries of habitat units, which, based on the NDVI, are clearly visible, and their accuracy, related to the resolution of satellite images on which the study was conducted ($10 \times 10$ m), is entirely sufficient for the needs of forestry practice.

The results of the research can also be used to monitor the forest with regard to possible changes in the trophicity of habitats, which should help to protect forest ecosystems in the conditions of changing water resources, as a result of the impact of natural and anthropogenic factors.

## 6. Patents

The article was created on the basis of the patent application PL P.439801 entitled "The method of determining the degree of forest moisture on the basis of remote sensing", the authors of which are: AM, MK, SK, PR, JP.

**Supplementary Materials:** The following supporting information can be downloaded at https://www.mdpi.com/article/10.3390/rs14174267/s1, Full: Ranking of indices, Index database used from https://www.indexdatabase.de/) [48].

**Author Contributions:** A.M., J.P., M.K. and P.R.: conceptualization and methodology; A.M.: investigation; A.M., S.K. and W.K.: formal analysis; A.M. and M.K.: visualization; A.M., M.K., S.K., P.R., J.P. and W.K.: writing original draft, review, and editing. A.M. and J.P.: funding acquisition; validation. All authors have read and agreed to the published version of the manuscript.

**Funding:** The research was financed from the authors' private funds.

**Data Availability Statement:** Restrictions apply to the availability of these data. Data was obtained from Bory Tucholskie National Park [BTNP] and are available from the authors with the permission of BTNP.

**Acknowledgments:** The authors thank Karolina Walczak for help in graphic processing and technical assistance.

**Conflicts of Interest:** The authors declare no conflict of interest.

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
