# Peer review of "Spectral Indices as a Tool to Assess the Moisture Status of Forest Habitats"

_remotesensing, doi:10.3390/rs14174267_

Round 1

Reviewer 1 Report

In my opinion, the results obtained in this paper are not sufficient to demonstrate that the scientific hypothesis formulated by the authors is verified. In other words, according to my point of view, the correlation identified between the various indices and the moisture content of forest habitats is not sufficient to demonstrate that through these spectral indices it is possible to estimate the variation in forest moisture. Therefore, further experiments and a more complete characterization of a methodology more appropriate to the objectives would be needed.

Author Response

We respect the Reviewer's comments, but apart from the general lack of acceptance for our work, it is difficult to find points in the review that could be directly addressed. The research we described was of a pilot nature. Therefore, we agree with the Reviewer's thesis that further experiments would be needed ”, but no one before us has conducted such research and has not shown the possibility of linking NDVI with the state of habitat moisture. We recognize this as a strong point of our study. Of course, we also intend to continue our research. Taking into account the pilot nature of our research, we tried to describe design and methods as precisely as possible. As the other two reviews show, design and methods are sufficiently described for the remaining two reviewers. Similarly, results and conclusions. Therefore, we will try to improve the work in accordance with the suggestions of the other two reviewers and we ask to accept our manuscript after our revision.

Reviewer 2 Report

In the abstract section, the author should include the quantitative results.

On page-1,6, the Author should write the full form of GIS, L2A

The author should describe the topography of the study area

The author should describe the NDVI, EVI, SAVI, SR520/670, OSAVI indexes, and FHMI indices with the formula used in this study. The following sources may be useful in the computation of NDVI/EVI: https://www.int-arch-photogramm-remote-sens-spatial-inf-sci.net/XLII-3-W6/535/2019/isprs-archives-XLII-3-W6-535-2019.pdf

Describe the Tools/software for data analysis, geometric and radiometric corrections were carried out by the authors.

Author Response

  1. In the abstract section, the author should include the quantitative results.

Authors: The quantitative results were included.

  1. On page-1,6, the Author should write the full form of GIS, L2A.

Authors: GIS was written in the full form and L2A was explained.

  1. The author should describe the topography of the study area.

Authors: Topography was described and Fig. 2c with topography of the study area was added.

  1. The author should describe the NDVI, EVI, SAVI, SR520/670, OSAVI indexes, and FHMI indices with the formula used in this study. The following sources may be useful in the computation of NDVI/EVI: https://www.int-arch-photogramm-remote-sens-spatial-inf-sci.net/XLII-3-W6/535/2019/isprs-archives-XLII-3-W6-535-2019.pdf

Authors: The source given by Rewiever was used and was useful (thanks to Reviewer) and the indices were described

  1. Describe the Tools/software for data analysis, geometric and radiometric corrections were carried out by the authors.

Authors: Software used in the study was described

Reviewer 3 Report

Review, Manuscript ID: remotesensing-1803354*

“Spectral Indices as a Tool to Assess the Moisture Status of Forest Habitats”

This paper compares and evaluates the practical use of spectral indices obtained from Sentinel-2 satellite images and the Forest Habitats Moisture Indices (traditionally used in the Polish forest habitats classification system) in forest management in the case of the Bory Tucholskie National Park (BTNP).

The paper is well structured and written and in the line with the scope of the journal of Remote Sensing. Methods are suitable to address the aim and objectives of the paper. 

I believe that the paper has an high potential for informing forest management decisions where decision makers need faster and cheaper solutions in large areas.

Author Response

The Authors thank the Reviewer for such positive opinion.

Reviewer 4 Report

The aim of the presented study was to find the relationship between the spectral indices obtained from satellite images and the Forest Habitats Moisture Indices used traditionally in the Polish forest habitats classification system.

I congratulate the authors, the paper is well written and shows a new methodology and an important results. But I listed bellow some minor suggestions:

L.19-22: pay attention, you have to write a short phrase and use "end point". You have in this sentence 4 lines and 2 "and"..."and to enable their practical use in forest management and the  protection of forest ecosystems throughout the all temperate zone. I suggest: "and to enable their practical use in forest management, as also, the  protection of forest ecosystems throughout the all temperate zone. Please correct.

L23-27: The hypothesis can be in the same sentence.

L249: y-axis = Please correct the number : x.xx or x.x? And exclude the title above the figure. The same suggestions for the others figures.

I would like to know if the authors find this can be use on a tropical forests too.

Author Response

Dear Reviewer

Thank you very much for your time and the valuable comments. It helped us to make our paper better. Below there is our answers to you comments

Reviewer 1 comments and suggestions for Authors:

L.19-22: pay attention, you have to write a short phrase and use "end point". You have in this sentence 4 lines and 2 "and"..."and to enable their practical use in forest management and the  protection of forest ecosystems throughout the all temperate zone. I suggest: "and to enable their practical use in forest management, as also, the  protection of forest ecosystems throughout the all temperate zone.

Authors’ response

The Reviewer rightly pointed out that the sentence is too long. So we have shortened the sentence. Especially that the removed content was similar to last sentence of the Abstract.

Reviewer 1 comments

L23-27: The hypothesis can be in the same sentence.

Authors’ response

The hypothesis was added

Reviewer 1 comments

L249: y-axis = Please correct the number : x.xx or x.x? And exclude the title above the figure. The same suggestions for the others figures.

Authors’ response

We have corrected the figures

Reviewer 1 comments

I would like to know if the authors find this can be use on a tropical forests too.

Authors’ response

We have hope that Yes. We have just testing it.

Reviewer 5 Report

Investigate the possibility to asses the variation in the humidity os forest habitats on the basis of spectral index is relevant, as a fast e low cost method for understending the water contend in forested areas.

I just suggest remove the figure from introduction.

Author Response

Dear Reviewer

Thank you very much for your valuable work and comments. Below is our answer to your suggestion.

Reviewer 2 comments and suggestions for Authors:

Investigate the possibility to assess the variation in the humidity of forest habitats on the basis of spectral index is relevant, as a fast and low cost method for understanding the water content in forested areas.

I just suggest remove the figure from introduction.

Authors’ response

We have removed the figure. We have consistently renumbered the figures in the text

Round 2

Reviewer 1 Report

Unfortunately I can only confirm the previous comment. I therefore confirm that in my opinion the high correlation between NDVI and the moisture content of forest habitats is not enough to be able to estimate the variation in forest humidity. It is acceptable to argue that this is preliminary data, however it is not an original study as there are several works on the subject of the correlation between spectral indices and groundwater (for example Pascoa et al., 2020; Barron et al., 2014; Gou et al., 2015).

Author Response

Dear Reviewer

Once again we would like to thank you for your valuable work and comments. We know the papers Pascoa et al., 2020; Barron et al., 2014; Gou et al., 2015, but the papers are focused on vegetation. Our topic is the ground water level assessed on the spectral indices and the humidity of forest habitats. The practical result of our study is the possibility of replacement of traditional method of assessing of forest habitat moisture the method based on satellite images. So we confirm our previous statement, that it is the preliminary study.

The results shown by Pascoa et al., 2020 are much more connected to the climate differentiation than the ground water table. We do not think, that the modeled deepest water table depth reaching in the research area 448 m, described in the Pascoa’s Methods has really impact on vegetation. Additionally, our data are based on much higher spatial resolution (100 m, not ≈1 km ) and our the results could be used in practice  in the resolution 100 × 100 m.

Barron et al., 2014 assumed “that due to limited precipitation over a dry period of six to seven months, soil moisture stores would be depleted, and the areas that maintain constant amount of green vegetation (greenness) and high level of surface moisture (wetness) were likely to have access to groundwater. The Authors correctly assumed that the vegetation, being green during the drought period  could has access to water, but did not prove it. The vegetation, being green during the drought period could use the air moisture coming from neighbored streams, rivers or other water reservoirs. We have based our study on the really recognition of presence of ground water on the different depth and shown that it is really relationship between ground water depth and the NDVI. Additionally we have shown that there, where the ground water is below 2.5 m, beyond the range of Polish trees root systems, more impoertant is the impact of neighborhood of lakes on habitat moisture.

Undoubtedly, the papers cited by the Reviewer are very valuable, so we referred to them in Discussion, but they do not affect significantly the results of our study. In general, all research based on Remote Sensing is important, so we believe that our paper also matters.